# Accuracy of Instrument Portal Placement Using a Custom-Made 3D-Printed Aiming Device versus Free Hand Technique in Canine Elbow Arthroscopy

**DOI:** 10.3390/ani13233592

**Published:** 2023-11-21

**Authors:** Piotr Trębacz, Jan Frymus, Anna Barteczko, Mateusz Pawlik, Aleksandra Kurkowska, Michał Czopowicz

**Affiliations:** 1Department of Surgery and Anaesthesiology of Small Animals, Institute of Veterinary Medicine, Warsaw University of Life Sciences-SGGW, Nowoursynowska 159 C Street, 02-776 Warsaw, Poland; 2Cabiomede Ltd., ul. Karola Olszewskiego 21, 25-663 Kielce, Poland; anna.barteczko@cabiomede.com (A.B.); mateusz.pawlik@cabiomede.com (M.P.); aleksandra.kurkowska@cabiomede.com (A.K.); 3Division of Veterinary Epidemiology and Economics Institute of Veterinary Medicine, Warsaw University of Life Sciences-SGGW, Nowoursynowska 159 C Street, 02-776 Warsaw, Poland; michal_czopowicz@sggw.edu.pl

**Keywords:** elbow arthroscopy, aiming device, 3D-printed prototype

## Abstract

**Simple Summary:**

Elbow arthroscopy is commonly performed in canine’s orthopedics. During this procedure, two portals are required: one to insert the arthroscope, and the second one for surgical instruments. Establishment of an instrument portal is challenging due to limited visibility (via the arthroscope), and little free space inside the joint. To make this procedure easier, the goal of this study was to create a 3D-printed prototype of a device aiming the needle preceding the portal, and to check its feasibility on 15 canine cadavers of different sizes and breeds. On each cadaver, the procedure was performed on both elbows—one using the prototype, and the second one with a free hand. The two techniques were compared according to the mean number of attempts needed to achieve an optimal position of the instrument portal. We conclude that the use of the prototype increases the likelihood of the needle guiding the portal into entering the joint properly during the first attempt, making the arthroscopy less traumatic.

**Abstract:**

While the insertion of the arthroscope into the elbow joint is relatively easy based on anatomical landmarks, obtaining a correctly located instrument portal is often difficult. Therefore, the goal of the study was to create a 3D-printed prototype of an aiming device for the guiding needle, and to check its feasibility. The study included fresh cadavers of 15 dogs, 9 males and 6 females, aged from 1 to 6 years (median 4 years) with body weight from 17 to 57 kg (median 30 kg). On each dog, we compared the number of attempts needed to obtain optimal direction of the guiding needle for the portal, using one elbow the prototype, and performing this as control on the opposite joint without the prototype (with a free hand). The number of attempts needed was significantly lower using the prototype (median 1) than on the control elbows (median 2, *p* = 0.009). The number of attempts was not correlated with the body weight neither in the case of experimental (R_s_ = 0.18, *p* = 0.532) nor control elbows (R_s_ = 0.13, *p* = 0.642). We conclude that the used prototype seems to be helpful in elbow joint arthroscopy.

## 1. Introduction

During arthroscopy, the location of the instrument portals is chosen according to the anatomy and the principles of triangulation, depending on where the arthroscope and arthroscopic instruments are to be introduced. The arthroscope and instruments have to form an acute-angled triangle. The base of this triangle is the skin, the arms are created by the arthroscope and an instrument, and the apex of the triangle is on the operative site [1,2]. The angle between the arthroscope and the instrument should be acute: 30–60° [3], 25–45° [4]. A lower angle makes the manipulation in the joint difficult due to collisions between the instruments and the arthroscope outside of the joint. On the other hand, an angle ≥ 90° makes the movements of the operator’s hands very difficult.

Elbow arthroscopy is commonly performed in canine orthopedics. This procedure is indicated in cases of forelimb lameness with elbow pain, crepitus, joint capsule distension, reduced range of motion, or radiographic changes compatible with elbow dysplasia, intra-articular fracture, etc. During arthroscopy, a single instrument portal is usually established caudal to the medial collateral ligament. A guiding needle is a tool which marks the correct location and subsequently determines the direction of the instrument insertion. Although the elbow joint is surrounded by a thin cuff of soft tissue, it can be difficult to correctly place the guiding needle into the joint, to mark the location and subsequently to determine and keep the direction of the instrument insertion. In addition, the telescopes mostly used for small animal arthroscopy have a viewing angle of 30° [5]. Such an angle causes a distortion of the image obtained, and further complicates the proper placing of the instrument. Finally, in patients with osteoarthritis and/or a history of elbow trauma, the pathological lesions may cause additional difficulties during this procedure. One solution to these problems can be the use of an arthroscopic aiming device, facilitating the correct placement of the guiding needle and thus the instrument portals [6].

To the best of the authors’ knowledge, there are only two reports in the veterinary literature describing the use of an arthroscopic guide to facilitate the creation of instrument portals in small animals. Only Lehman and Lehman [2] and Riener et al. [7] have reported very good results of the use of an arthroscopic aiming device facilitating the creation of instrument portals during shoulder arthroscopy in dogs.

As there is not enough experience in using arthroscopic aiming devices in canine elbow surgery, the goal of this study was to check the feasibility of a custom-made, 3D-printed prototype of such an instrument.

## 2. Materials and Methods

### 2.1. 3D Design and Printing

The design and conversion of the arthroscopic aiming device into a 3D computer prototype involved the following steps:-Converting the CAD (computer aided design) files into 3D surface models consisting of triangle meshes to create a STL (stereolithography) file.-Processing the model with software to reduce noise and artefacts, and converting the geometry file to a machine code file type recognized by the 3D printer.

The angle between the arthroscope (Arthrex, 1.9 mm telescope, 30° viewing angle, ⌀ 3 mm arthroscopic sheath, working length 42 mm, Munich, Germany) and the guiding needle was set at 35°. Based on the experience of the authors, such an angle proved to be optimal for convenient interarticular manipulation with the instruments. A snap connection was chosen as the method of connecting the aiming device to the arthroscopic sheath. The dimensions of the guide depend on the length of the working part of the telescope cannula and the angle previously assumed between the arthroscope and the guide needle (Figure 1). The virtual model of the prototype was saved in the STL (stereolithography) format. An Ultimaker 3 printer was then used to print the device with predetermined parameters. This was performed using fused deposition modelling (FDM) technology, a technique in which successive, very thin layers of semi-liquid material are applied to the object being created from a heated printer nozzle. Carbon fiber-reinforced nylon was used to print the prototype, then the supports were removed and the surface was sandblasted, giving a finalized product (Figure 2).

### 2.2. Cadaver Studies

All arthroscopies were performed on 15 fresh cadavers including 5 crossbreed and 10 pedigree dogs. Their breeds, sex and age are given in Table 1. The cadavers were positioned in lateral recumbency with the limb having arthroscopy downward, which allowed the use of the edge of table as a fulcrum for distraction of the joint. After shaving prepping and draping of the elbow area, the joint was filled with 0.9% NaCl. The arthroscope was inserted below the medial humeral epicondyle, just behind the superficial flexor muscle. The medial coronoid process and the medial collateral ligament were then visualized on the video monitor. Next, with the arthroscope kept still, a 1.2 × 88 mm spinal needle (KD-Fine, Berlin, Germany) was inserted caudally to the medial collateral ligament. In each cadaver, the needle was randomly (by a coin toss) inserted, fixed in the aiming prototype (experimental joint) in one of the elbows (Figure 3), and without support (free hand) on the other site (control joint). In case of free hand introduction, the needle was inserted caudally to the medial collateral ligament, cranially and slightly proximally to the telescope portal. The prototype was used on 8 left and 7 right elbow joints. For both methods of guiding needle insertion, insertion was considered correct if the tip of the needle could be seen in the center of the field of view of the arthroscope. In case of failure, the needle was re-inserted until proper position was obtained. The number of attempts needed to obtain a proper position of the guiding needle in the elbow joint was analyzed. After accurate insertion of the needle, the instrument portal was created. A stab incision was made in the joint cavity, at the site of needle insertion, using blade no. 11. If the aiming device was used, after releasing the snap connection between the arthroscopic sheet and the aiming device, the device was moved away from the skin surface without retrieving the guiding needle. As described above, a bayonet scalpel blade was used to create the instrumental portal (Figure 4). No additional instruments were inserted into the elbow joint once the instrument portal was created.

Additionally, in order to assess the resistance of the material against high temperature, the prototype was steam-sterilized at 121 °C (Melag 23 autoclave, Berlin, Germany) for 15 min each time before use, that is, 15 times in total.

### 2.3. Statistical Analysis

The number of attempts was presented as the median and range. It was compared between sides and between experimental and control elbows using the Wilcoxon signed rank test. Correlation between the number of attempts and the body weight was examined using Spearman’s rank correlation coefficient (R_s_). A two-tailed significance level (α) was set at 0.05. The analysis was performed in TIBCO Statistica 13.3 (TIBCO Software Inc., Palo Alto, CA, USA).

## 3. Results

### 3.1. 3D Design and Printing

It took two working days to design, print and finish the prototype. Despite autoclaving fifteen times, no negative effect of sterilization was noticed on this device.

### 3.2. Cadaver Studies

The number of attempts to place the guiding needle properly ranged from 1 to 4 (median of 1). The number of attempts did not differ significantly between the left and the right elbow (*p* = 0.625). The number of attempts was significantly lower on experimental than on control elbow joints (*p* = 0.009) (Table 1, Figure 5). The number of attempts was not correlated with the body weight either in the case of experimental (R_s_ = 0.18, *p* = 0.532) or control elbow joints (R_s_ = 0.13, *p* = 0.642).

## 4. Discussion

In dogs, elbow arthroscopy is mostly performed for elbow dysplasia surgery. In this case, a single instrument portal is usually created caudally to the medial collateral ligament and cranial to the arthroscope. The subsequent comfort of the surgical instruments working inside the joint is directly influenced by the location of the portal. In general, the needle used to determine the location of the instrument portal is inserted manually perpendicular to the skin surface without the use of additional instruments. The most common reason for failure, when inserting the needle this way is inserting it to close to the arthroscope, or in a wrong angle, so that the arthroscope is missed by the tip of the needle [3,5]. In result, it has been estimated that the learning time for proper performing of this procedure is about one year, and that there are about 30 approaches needed [2]. Each attempt can damage the joint tissues. Iatrogenic articular cartilage injury is a known complication of arthroscopy. There is a critical need to prevent iatrogenic articular cartilage injury because canine articular cartilage has a limited healing capacity [2,6].

In our study, we compared the effectiveness of two techniques for inserting the needle into the elbow joint in the cadavers of 15 dogs of different size and breeds. All procedures were performed by a surgeon with approximately 5 years’ experience of performing arthroscopic surgery (P.T.). We can conclude that the use of a custom-made aiming device increases the likelihood of the needle entering the joint properly at the first attempt. Similarly, Lehman and Lehman [2] found that the use of an arthroscopic aiming device allowed easier, quicker and less traumatic insertion of the working cannula into the caudal joint pouch of the canine shoulder joint.

Riener et al. [8] found that the use of an arthroscopic aiming device allowed more successful insertion of the cranial working cannula into the shoulder joint. This was achieved more quickly and was associated with fewer insertion attempts and less damage to the articular cartilage. By using the same device in all cadavers, we demonstrated that it can be helpful regardless of the size of the dog as the number of attempts was not correlated with the body weight either in the case of experimental (*p* = 0.532) or control elbows (*p* = 0.642).

When analyzing the results of 15 needle insertions without the support of the aiming device, the results were unsatisfactory. In only five joints was it possible to insert the needle correctly at the first trial, and in two cases as many as four attempts were required (Table 1). This was probably due to technical errors like choosing a wrong insertion point and/or angle or placing the needle outside the joint cavity. The needle should be inserted perpendicular to the skin surface and in relation to the arthroscope. Difficulties in positioning the needle for insertion without additional instruments may be related to some distortion of the image displayed on the video monitor associated with the use of a 30° telescope.

As mentioned above, the angle between the arthroscope and the guiding needle is crucial for comfortable manipulation with the instruments inside the joint. As we did not find any information in the literature on what is the optimal angle, we chose 35°. According to our experience, such an angle allows for the comfortable use of arthroscopic instruments without collision with the arthroscope. In this study, we did not compare the ability and comfort of using arthroscopic instruments inserted into the joint through the instrument portals created after insertion of the guide needle without the use of additional instruments and with the aiming device. This issue requires further investigation.

Using the CAD/CAM (computer-aided design/computer-aided manufacturing) technology and 3D printing, it was possible to produce a functional arthroscopic guide in a short time. Prototyping and 3D printing offer virtually unlimited possibilities to create new instruments in a short time. Currently, guides for orthopedic drills, bone saw guides, anatomical models for intra-operative navigation, etc., are created in this way [8,9,10,11,12]. FDM is a technique in which successive, very thin layers of semi-liquid material are applied to the object being created from a heated printer nozzle, using nylon with the addition of carbon fibers. The material used to make medical instruments must ensure that it retains adequate strength properties while allowing the device to be steam sterilized at 121 °C. Carbon fiber-reinforced nylon used in our study is highly temperature resistant, and yet elastic enough to make it possible to use the snap-fit method of connecting the device to the arthroscope. We confirmed that after 15 steam sterilizations no noticeable changes of this prototype were seen. In addition, in the case of our guide, prototyping and printing took only two days. Furthermore, the simple design of the aiming guide makes it easy to modify the prototype and print instruments to fit other joints.

The main limitation of this study is the comparison of the techniques of the guiding needle introduction instead of an instrument port. However, the instrument port location usually corresponds with the guiding needle position, as commonly a bayonet scalpel blade inserted along the guiding needle makes a tunnel for the instrument portal [5]. Also, the cadaveric nature of the study could be a limitation, as post-mortem changes can have influenced the elasticity of the elbow joint that may not fully reflect tissue flexibility in living animals. Furthermore, another limitation of our study is that we did not assess and compare the time taken to place the guide needle correctly using the two methods described. We did attempt to measure the time, but we realized that time was not correlated with needle insertion but with the number of attempts to achieve the correct position. With this in mind, we focused on counting the number of needle insertions as a more objective parameter. 

In conclusion, the evaluated prototype appears to be useful instrument for elbow arthroscopy, especially for untrained surgeons, making the whole procedure less traumatic. However, we believe that experienced operators can also benefit from this device. The triangulation provided by the printed aiming device ensures that the tip of the arthroscope and the instrument are properly aligned for optimal function.

## Figures and Tables

**Figure 1 animals-13-03592-f001:**
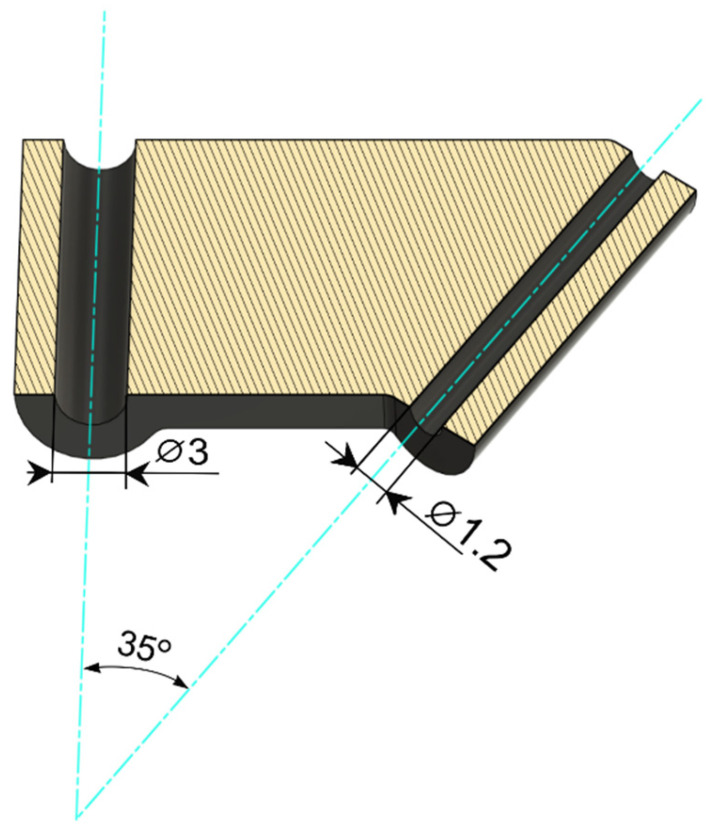
The virtual model of the custom-made arthroscopic aiming device (transverse section).

**Figure 2 animals-13-03592-f002:**
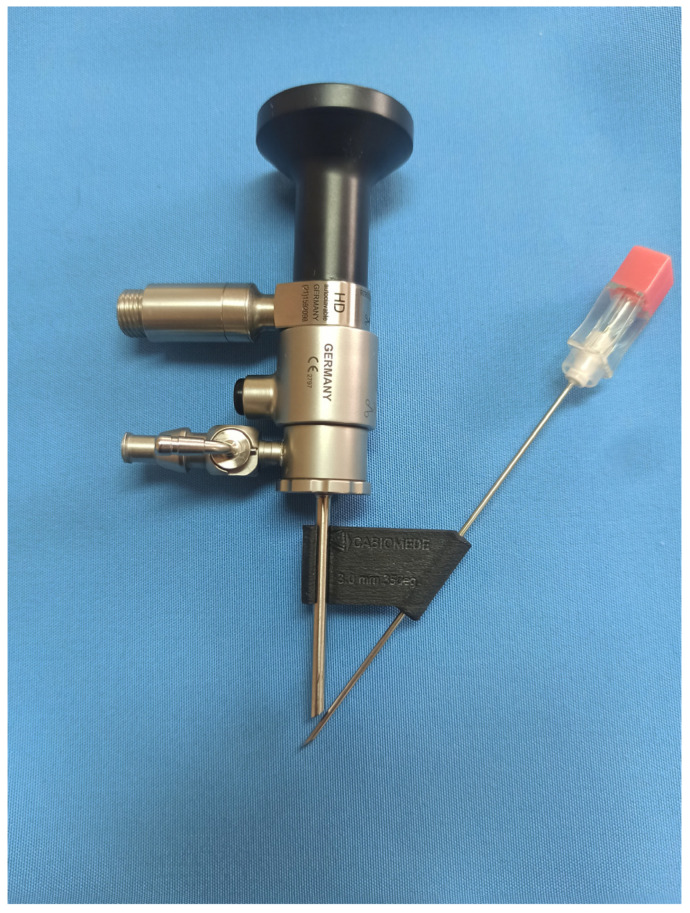
Arthroscope, custom-made aiming device and spinal needle ready to use.

**Figure 3 animals-13-03592-f003:**
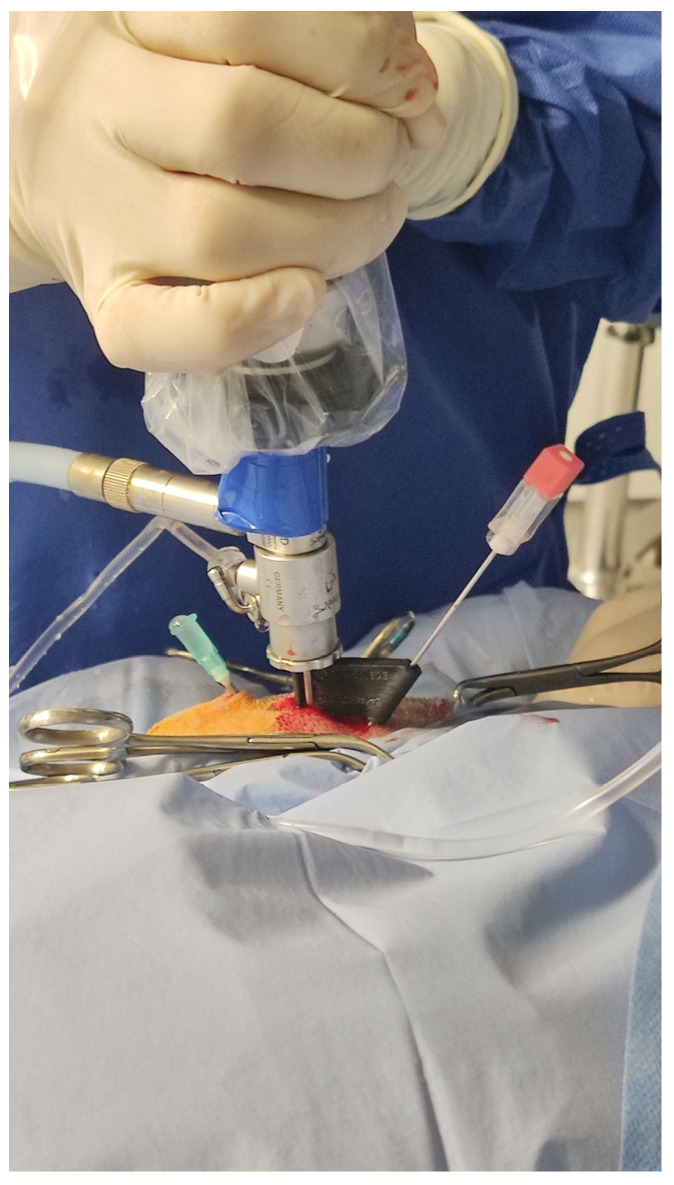
Insertion of the needle with assistance of the aiming device in dog no. 1 (head on the right).

**Figure 4 animals-13-03592-f004:**
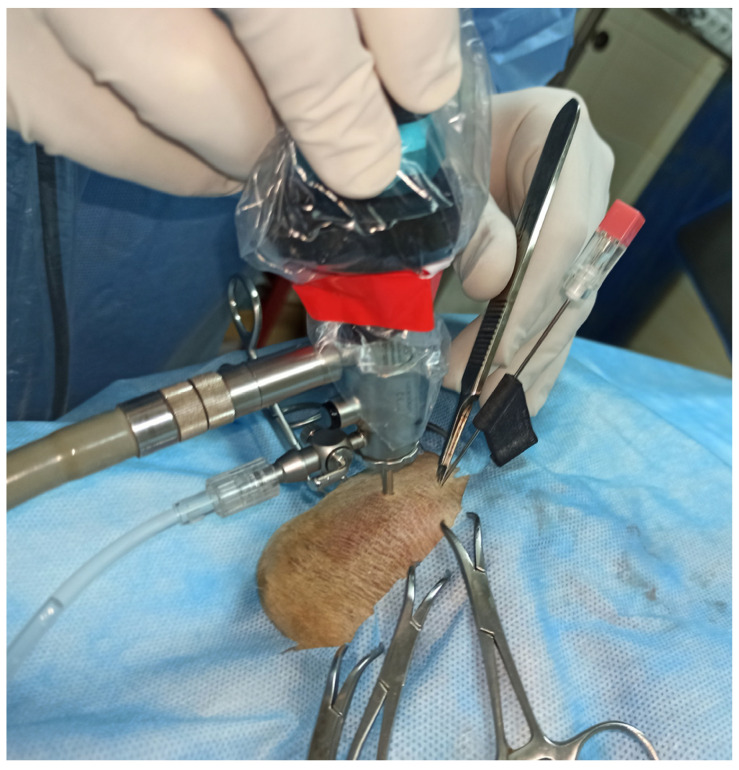
Creation of the instrument portal after releasing the snap connection between the arthroscopic sheet and the aiming device in dog no. 9 (head to the right).

**Figure 5 animals-13-03592-f005:**
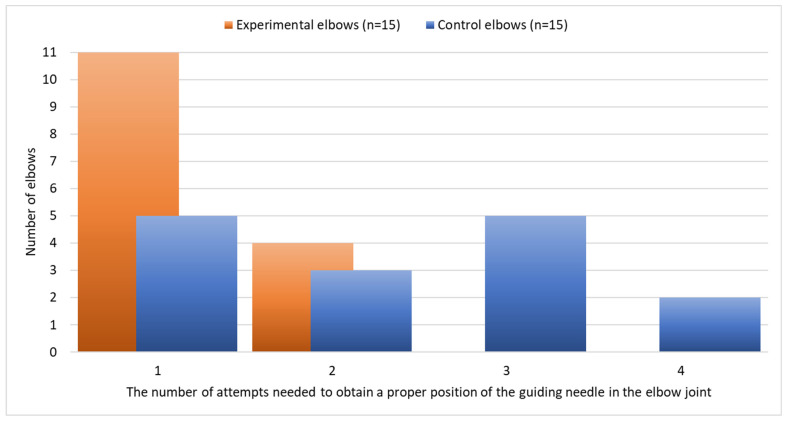
Number of attempts needed to obtain a proper position of the guiding needle in the elbow joint on experimental and control elbows.

**Table 1 animals-13-03592-t001:** Number of attempts needed to obtain proper position of the guiding needle on experimental (signed *) and control elbows and descriptive statistical analysis.

No.	Cadaver Characteristics				Elbow		Number of Attempts Needed to Place the Guiding Needle Properly	
	Breed	Sex	Age (Years)	Body Weight (kg)	Left	Right	Custom-Made Arthroscopic Aiming Device (Experimental Elbow)	Free Hand (Control Elbow)
1	Crossbreed	male	5	31	1	1 *	1	1
2	Boerboel	male	6	57	1 *	2	1	2
3	Newfoundland	male	2	45	1 *	4	1	4
4	Labrador retriever	male	2	29	3	1 *	1	3
5	Crossbreed	male	2	19	1	1 *	1	1
6	German Shepherd	male	3	35	2 *	1	2	1
7	Chow-chow	female	1	17	1 *	3	1	3
8	Crossbreed	female	6	35	1	1 *	1	1
9	Golden retriever	female	4	32	2	1 *	1	2
10	Crossbreed	male	3	25	3	2 *	2	3
11	Labrador retriever	male	6	35	2 *	4	2	4
12	Staffordshire Bull Terrier	male	2	17	1 *	1	1	1
13	Crossbreed	female	5	20	1 *	2	1	2
14	Labrador retriever	female	4	30	2 *	3	2	3
15	Staffordshire Bull Terrier	female	6	27	3	1 *	1	3
	**Descriptive statistics**							
	Median		4	30	1	1	1	2
	Range		1	17	1–3	1–4	1–2	1–4
	*p*-value		6	57	0.625		0.009	

## Data Availability

Data are contained within the article.

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
