# Peer review of "Accuracy of Instrument Portal Placement Using a Custom-Made 3D-Printed Aiming Device versus Free Hand Technique in Canine Elbow Arthroscopy"

_animals, 2023, doi:10.3390/ani13233592_

Round 1

Reviewer 1 Report

Comments and Suggestions for Authors

Simple summary: line 14 improve sentence structure

Abstract: line 25 "was THE creation"

Introduction: it would be interesting to describe why elbow arthroscopy is performed in the dog. 

Materials and methods: Describe in more detail the technical construction of the instrument and why the chosen angle was 35°.

Results: poorly detailed 

The sterilisation technique and the number of times it was performed should be described in materials and methods. It cannot be mentioned in the results section alone. 

Improve the English language. 

Discussion: Add technique and results of studies describing the method for shoulder arthroscopy. 

Comments on the Quality of English Language

The article needs to improve its syntax. Periods are often very basic and not suitable for scientific language. Periods are often very basic and not suitable for scientific language 

Author Response

At firs many thanks for the critical review and valuable suggestions. They will definitely improve our paper.

Below the modifications of the manuscript:

Simple summary: line 14 improve sentence structure

Answer: we improved

Abstract: line 25 "was THE creation

Answer: we improved

Introduction: it would be interesting to describe why elbow arthroscopy is performed in the dog. 

Answer:

We add Line 54 “This procedure is indicated in cases of forelimb lameness with elbow pain, crepitus, joint capsule distension, reduced range of motion, or radiographic changes compatible with elbow dysplasia, intra-articular fracture, etc.”

Materials and methods: Describe in more detail the technical construction of the instrument and why the chosen angle was 35°.

Answer:

We add Line 80 „The design and conversion of the arthroscopic aiming device into a 3D computer prototype involved the following steps:

- converting the CAD (computer aided design) files into 3D surface models consisting of triangle meshes to create a STL (stereolithography) file.

- processing the model with software to reduce noise and artefacts, and converting the geometry file to a machine code file type recognized by the 3D printer.”

We add line 92“The dimensions of the guide depend on the length of the working part of the telescope cannula and the angle previously assumed between the arthroscope and the guide needle.”

We add Line  97 , a technique in which successive, very thin layers of semi-liquid material are applied to the object being created from a heated printer nozzle”

and why the chosen angle was 35°.

Answer:

We add line 89 “Based on the experience of the authors, such an angle  proved to be optimal for convenient interarticular manipulation with the instruments.”

We add Line 210 As we did not find any information in the literature what is the optimal angle, we choose 35°. According to our experience, such an angle allows the comfortable use of arthroscopic instruments without collision with the arthroscope.”

The sterilisation technique and the number of times it was performed should be described in materials and methods. It cannot be mentioned in the results section alone. 

Answer:

We add  Line 131 In order to assess the resistance of the material against high temperature, the prototype was steam sterilized at 121ºC (Melag 23 autoclave, Germany) for 15 minutes each time before use, that is 15 times in total.”

Discussion: Add technique and results of studies describing the method for shoulder arthroscopy. 

Answer:

We add line 193 “We can conclude that the use of the a custom-made aiming device increases the likelihood that the needle will enter the joint properly at the first attempt. Similar observations were made by Lehman and Lehman [2] and Riener et al. [6] who investigated the usefulness of an aiming device in creating instrument portals during canine shoulder arthroscopy. Similarly, Lehman and Lehman (2) found that the use of an arthroscopic aiming device allowed easier, quicker and less traumatic insertion of the working cannula into the caudal joint pouch of the canine shoulder joint.

Riener et al. (6) found that the use of an arthroscopic aiming device allowed more successful insertion of the cranial working cannula into the shoulder joint. This was achieved more quickly and was associated with fewer insertion attempts and less damage to the articular cartilage.”

Reviewer 2 Report

Comments and Suggestions for Authors

I have read and reviewed this manuscript with great interest and overall, from this reviewer's perspective, it is a study that has been well-planned and executed. Overall it is a study with refreshingly simple wording that is easy to understand. The results are clearly presented. The literature is cited correctly and appropriately to the topics covered.

However, a clarification needs to be addressed to achieve publication quality. I have left some comments hoping that they will help the authors.

Was the self-made targeting device the same size for each dog?

Because you mentioned different breeds and different sizes.

Please clarify more clearly in the manuscript.

Author Response

Dear Reviewer,

Many thanks again for your suggestions.

We incorporated all of them:

Below the modifications of the manuscript:

Was the self-made targeting device the same size for each dog?

Because you mentioned different breeds and different sizes.

Answer: Yes, we use the same size aiming device for all cadaver. We found that the number of attempts did not correlate with body weight in either experimental (Rs = 0.18, p=0.532) or control elbows (Rs=0.13, p=0.642).

Please clarify more clearly in the manuscript.

Answer:

We add  line 203 we add “ Using the same device in all cadavers we demonstrated that it can be helpful regardless of the size of the dog as the number of attempts was not correlated with the body weight either in the case of experimental (p=0.532) or control elbows (p=0.642).”

Reviewer 3 Report

Comments and Suggestions for Authors

The authors describe a printed device for improving the elbow arthroscopy in dogs. The manuscript appears well written, scientifically sounds and this device could be of interest in surgical procedures. It could be useful in other species too, as in other joints. The major issues are:

1) the description of the printing method to realize the device can be improved.

2) the authors shoud better explicate why the angle of 35° is the better solution. 

3) from a geometrical point of view, the same angle of 35° can hard to be employed in different elbow joints of different dog breeds. The device could be to have a variable angle to better applied the device in different dogs. 

4) some other images are required

After this the article deserves to be published. 

Author Response

Many thanks for very valuable remarks.

Below our answers:

the description of the printing method to realize the device can be improved.

Answer:

We add Line 80 „The design and conversion of the arthroscopic aiming device into a 3D computer prototype involved the following steps:

- converting the CAD (computer aided design) files into 3D surface models consisting of triangle meshes to create a STL (stereolithography) file.

- processing the model with software to reduce noise and artefacts, and converting the geometry file to a machine code file type recognized by the 3D printer.”

We add line 92 “The dimensions of the guide depend on the length of the working part of the telescope cannula and the angle previously assumed between the arthroscope and the guide needle.”

We add Line  96 , a technique in which successive, very thin layers of semi-liquid material are applied to the object being created from a heated printer nozzle”

 the authors shoud better explicate why the angle of 35° is the better solution. 

Answer:

We add line 89 “Based on the experience of the authors, such an angle  proved to be optimal for convenient interarticular manipulation with the instruments.”

We add Line 219 As we did not find any information in the literature what is the optimal angle, we choose 35°. According to our experience, such an angle allows the comfortable use of arthroscopic instruments without collision with the arthroscope.”

from a geometrical point of view, the same angle of 35° can hard to be employed in different elbow joints of different dog breeds. The device could be to have a variable angle to better applied the device in different dogs. 

We have used the same aiming device for all cadavers, and we found that the number of attempts did not correlate with body weight in either experimental (Rs = 0.18, p=0.532) or control elbows (Rs=0.13, p=0.642). However an adjustable aiming device may be more useful for different joints. Thank you for your suggestion. It would be valuable to develop such a device.

We add line 203 “Using the same device in all cadavers we demonstrated that it can be helpful regardless of the size of the dog as the number of attempts was not correlated with the body weight either in the case of experimental (p=0.532) or control elbows (p=0.642).

 some other images are required

We add Figure 2. Virtual model of the aiming device.

Round 2

Reviewer 3 Report

Comments and Suggestions for Authors

The manuscript has been improved. All the suggestions are taken in account and then it can be published. 

Author Response

Thank you